# *Trichoderma virens* and *Pseudomonas chlororaphis* Differentially Regulate Maize Resistance to Anthracnose Leaf Blight and Insect Herbivores When Grown in Sterile versus Non-Sterile Soils

**DOI:** 10.3390/plants13091240

**Published:** 2024-04-30

**Authors:** Pei-Cheng Huang, Peiguo Yuan, John M. Grunseich, James Taylor, Eric-Olivier Tiénébo, Elizabeth A. Pierson, Julio S. Bernal, Charles M. Kenerley, Michael V. Kolomiets

**Affiliations:** 1Department of Plant Pathology and Microbiology, Texas A&M University, College Station, TX 77843-2132, USA; pei-cheng.huang@ag.tamu.edu (P.-C.H.); peiguo.yuan@ag.tamu.edu (P.Y.); jimtaylor011@gmail.com (J.T.); elizabeth.pierson@ag.tamu.edu (E.A.P.); c-kenerley@tamu.edu (C.M.K.); 2Department of Entomology, Texas A&M University, College Station, TX 77843-2475, USA; johngrunseich@tamu.edu (J.M.G.); julio.bernal@ag.tamu.edu (J.S.B.); 3Department of Horticultural Sciences, Texas A&M University, College Station, TX 77843-2133, USA; eric.tienebo@inphb.ci; 4Agronomic Sciences and Transformation Processes Joint Research and Innovation Unit, Institut National Polytechnique Félix Houphouët-Boigny, Yamoussoukro P.O. Box 1093, Côte d’Ivoire

**Keywords:** *Trichoderma virens*, *Pseudomonas chlororaphis*, induced systemic resistance, jasmonic acid, *Spodoptera frugiperda*, *Diabrotica virgifera virgifera*

## Abstract

Soil-borne *Trichoderma* spp. have been extensively studied for their biocontrol activities against pathogens and growth promotion ability in plants. However, the beneficial effect of *Trichoderma* on inducing resistance against insect herbivores has been underexplored. Among diverse *Trichoderma* species, consistent with previous reports, we showed that root colonization by *T. virens* triggered induced systemic resistance (ISR) to the leaf-infecting hemibiotrophic fungal pathogens *Colletotrichum graminicola*. Whether *T. virens* induces ISR to insect pests has not been tested before. In this study, we investigated whether *T. virens* affects jasmonic acid (JA) biosynthesis and defense against fall armyworm (FAW) and western corn rootworm (WCR). Unexpectedly, the results showed that *T. virens* colonization of maize seedlings grown in autoclaved soil suppressed wound-induced production of JA, resulting in reduced resistance to FAW. Similarly, the bacterial endophyte *Pseudomonas chlororaphis* 30-84 was found to suppress systemic resistance to FAW due to reduced JA. Further comparative analyses of the systemic effects of these endophytes when applied in sterile or non-sterile field soil showed that both *T. virens* and *P. chlororaphis* 30-84 triggered ISR against C. *graminicola* in both soil conditions, but only suppressed JA production and resistance to FAW in sterile soil, while no significant impact was observed when applied in non-sterile soil. In contrast to the effect on FAW defense, *T. virens* colonization of maize roots suppressed WCR larvae survival and weight gain. This is the first report suggesting the potential role of *T. virens* as a biocontrol agent against WCR.

## 1. Introduction

Maize is the top-yielding cereal crop produced globally and a major source of livestock feed, biofuel, and staple food for many regions worldwide [1]. Pathogen infection and insect herbivory are constant threats to crop production, causing over 20% yield losses of maize annually on a global level and resulting in substantial economic impact and food insecurity [2]. The hemibiotrophic fungal pathogen *Colletotrichum graminicola* is the causal agent of anthracnose leaf blight and stalk rot in maize and accounts for millions of dollars in yield losses annually in the United States [3,4,5]. Fall armyworm (FAW; *Spodoptera frugiperda*) originates in the Americas and has recently invaded Europe, Africa, Asia, and Australia, causing serious damage [6,7]. FAW larvae feed on all of the aboveground tissues of maize, resulting in several billions of dollars in yield losses globally [6]. Western corn rootworm (WCR; *Diabrotica virgifera virgifera*) is the most economically important maize pest in the United States, and causes losses in excess of US $1 billion annually in the United States and Europe [8,9,10]. WCR larvae cause the most significant damage through feeding on the maize root, while WCR adult beetles damage the silks [8,11]. Published evidence indicates that the plant phytohormone salicylic acid (SA) plays a critical role in maize defense against *C. graminicola* [12,13], while resistance to chewing insects, including FAW, is mainly mediated by the lipid-derived phytohormone jasmonic acid (JA) [14,15,16]. However, resistance mechanisms against root-feeding WCR remain largely unknown [17].

Microbial biological control agents, including plant growth-promoting rhizobacteria and fungi (PGPR and PGPF, respectively), have been widely used to increase resistance in plants against pathogens and pests [18,19,20]. Isolates of soil-borne *Trichoderma* spp. And the bacterium *Pseudomonas chlororaphis* are well-known biocontrol agents and have been extensively studied for their beneficial traits, including growth promotion and enhanced disease resistance in host plants [21,22,23,24]. However, using *Trichoderma* spp. as biocontrol agents against pests has only recently been considered, and their impact on defense against chewing insects remains largely unexplored [25,26]. Several studies have shown that *T. virens* induces systemic resistance (ISR) against leaf-infecting pathogens, the physiological process of which is tightly regulated by the fungal small secreted proteins Sm1 and Sir1, which play pivotal but contrasting roles in triggering ISR responses [27,28,29,30]. Analyses of *T. virens sm1* deletion mutants have revealed that they are unable to induce ISR against fungal pathogens in above-ground organs, suggesting that the Sm1 peptide is a positive regulator of ISR [27,28]. In sharp contrast, deletion of *SIR1* resulted in elevated *SM1* expression [30] and a much stronger ISR response [29]. Wang et al. (2020) [30] showed that *T. virens* colonization in maize roots upregulated genes involved in the biosynthesis of JA precursors, including 12-OPDA (12-oxo-10(Z),15(Z)-phytodienoic acid) and OPC-4:0 {(*Z*)-4-[3-oxo-2-(pent-2-en-1-yl)cyclopentyl]butanoic acid}, yet downregulated genes downstream of 12-OPDA for JA biosynthesis. Whether *T. virens* colonization affects wound-induced JA production in leaves and confers resistance to leaf-feeding FAW and root-feeding WCR larvae remains unexplored.

The original objective of this study was to investigate the effects of *T. virens* and its secreted peptide signals on the production of wound-induced JA and other defensive oxylipin metabolites, and to test whether the beneficial effects of this endophyte extend to increasing resistance against FAW. Contrary to our expectation, the results demonstrated that maize seedlings grown in sterile soil amended with wild-type *T. virens* (TvWT), specifically *sm1* or *sir1* single or double mutants, were found to exhibit suppressed wound-induced JA biosynthesis, resulting in reduced resistance to FAW. Such unexpected results prompted us to further test whether such a detrimental effect on defense against herbivory could be ascribed to another well-studied growth-promoting bacterial endophyte, *P. chlororaphis*. Colonization of roots by the *P. chlororaphis* 30-84 also reduced production of JA and defense against FAW in sterile soil. These results necessitated the next objective, which was to test whether reduced defense against herbivory could also be observed in plants grown under non-sterile soil conditions. The results showed no detrimental change in resistance to FAW. Such a differential impact of soil sterility prompted us to test whether soil conditions alter the effectiveness of ISR triggered by both biocontrol agents against the leaf pathogen *C. graminicola*, and the results showed strong induction of ISR regardless of soil condition. The final objective of this study was to test whether *T. virens* treatment impacts maize interactions with root-feeding WCR larvae. We showed that colonization of roots by *T. virens* reduced larvae survival and growth.

## 2. Results

### 2.1. Colonization of Maize Roots by Either Δsir1 or Δsm1Δsir1 Enhanced Greater Levels of ISR against Anthracnose Leaf Blight Caused by C. graminicola

While our previous research characterized the contrasting functions of the secreted peptides, Sm1 and Sir1, in the regulation of ISR against pathogens, the hypothesized antagonistic interaction between the two signaling peptides has not been tested previously. Wang et al. (2020) [30] showed that the expression of *SIR1* in *Δsm1* remained similar to the level seen in the wild-type *T. virens* strain (TvWT), while *Δsir1* had elevated expression of *SM1*. To test whether increased ISR by *Δsir1* is due to elevated *SM1* expression in this strain, we created a *Δsm1Δsir1* double mutant and tested its capability to induce resistance to *C. graminicola*. The results showed that colonization by both *Δsir1* and *Δsm1Δsir1* mutants triggered a greater level of ISR against *C. graminicola* compared to TvWT (32%, 59%, and 56% reduction in lesion areas on TvWT-, *Δsir1*-, and *Δsm1Δsir1*-treated plants, respectively, compared to control) (Figure 1). These results suggest that functional Sir1 suppresses disease resistance to *C. graminicola* by not only suppressing *SM1*, but also through other as yet unknown mechanisms.

### 2.2. T. virens Colonization Suppressed Insect Defense

Because the effect of *T. virens* on ISR against insect herbivores has not been explored before, we tested whether TvWT and the secreted peptide signaling mutants *Δsm1*, *Δsir1*, and *Δsm1Δsir1* induce differential resistance to the foliar-feeding FAW. Surprisingly, the results showed that FAW larvae gained significantly more weight after feeding on *T. virens*-colonized plants for 7 days (60% to 95% increase) (Figure 2A), and third-instar larvae consumed more leaf area on *T. virens*-colonized plants (64% to 83% increase) (Figure 2B). Treatment with *T. virens* reduced resistance to FAW, regardless of the mutation of these two signaling peptides.

### 2.3. T. virens Treatment Reduced Wound-Induced JA Accumulation by Suppressing Biosynthesis and Enhancing Its Catabolism

Because *T. virens* treatment decreased insect defense, and JA dominantly modulates defense responses against chewing insects [14], we tested whether *T. virens* colonization regulates wound-induced JA production in leaves. Phytohormone and oxylipin profiling showed that significantly lower amounts of jasmonates, including JA precursors, 12-OPDA, OPC-4:0, and JA, and the bioactive JA derivatives, JA-isoleucine (JA-Ile), JA-leucine (JA-Leu), and JA-valine (JA-Val) conjugates, were accumulated in the *T. virens*-treated plants at 1 and 2 h post wounding (hpw) (Figure 3A). Specifically, JA accumulation was reduced by 23% to 37% and 33% to 38% and JA-Ile content was reduced by 8% to 37% and 44% to 50% compared to the amounts accumulated in the control plants at 1 and 2 hpw, respectively (Figure 3B,C). Notably, we also found that one of the JA catabolites, 12OH-JA, accumulated to a significantly higher level in the *T. virens*-treated plants in response to mechanical wounding (83% to 171% and 23% to 82% increase compared to the contents in the control plants at 1 and 2 hpw, respectively) (Figure 3A,D). This latter result suggests that one potential reason behind decreased JA content may be its increased catabolism. To confirm that the reduced insect defense by *T. virens* colonization was modulated by JA signaling, we tested the effect of *T. virens* on resistance to FAW in the JA-deficient *opr7opr8* mutant [14]. The result showed that *T. virens* colonization did not increase susceptibility to FAW in *opr7opr8* double mutants, suggesting that attenuated JA signaling plays an essential role in *T. virens*-mediated suppression of defense against FAW (Figure 4). Interestingly, the levels of the α-ketol 9,10-KODA (9-hydroxy-10-oxo-12(Z),15(Z)-octadecadienoic acid) were significantly reduced at 1 hpw (24% to 37% reduction compared to the contents in the control plants) by all strains of *T. virens* (Figure 3E). This molecule has been recently shown to play a major JA-dependent signaling role in defense against FAW [31]. Together, these results indicate that *T. virens* colonization suppressed defense against FAW via downregulating JA biosynthesis and/or enhancing JA catabolism and its downstream ketol biosynthesis.

### 2.4. T. virens and P. chlororaphis Colonization Induced Resistance to C. graminicola in Both Sterile and Non-Sterile Soil Conditions, but Enhanced Growth and Suppressed Insect Defense against FAW Only in Sterile Soil

Because of the detrimental effect of the growth-promoting *T. virens* on herbivory resistance, we next tested whether another growth-promoting microorganism would have a similar impact on defense. For this, we chose the bacterial endophyte *P. chlororaphis* 30-84, which has been reported to promote plant growth and suppress the growth of fungal pathogens [32,33]; however, its role in insect defense against foliar-feeding insects has remained mostly unexplored. Specifically, we tested the impact of this bacterium on plant growth, defense against FAW, and pathogen resistance, and the effect of combining *T. virens* and *P. chlororaphis* on these physiological processes. Because beneficial microorganisms are commonly tested in agriculturally-relevant settings, we further tested their effect in both sterile and non-sterile soil mixtures of field soil and potting mix. Similar to *T. virens* treatment, colonization by *P. chlororaphis* 30-84 promoted plant growth (12% to 15% increase in plant height), but suppressed defense against FAW (41% to 51% increase in larvae weight gain) in the sterile soil mixture (Figure 5A,B). Plant growth and insect defense were not significantly affected when seedlings were grown in a non-sterile soil mixture upon treatment with either *T. virens* or *P. chlororaphis* 30-84. However, both *T. virens* and *P. chlororaphis* were able to increase resistance to *C. graminicola* in both soil types (53% to 66% reduction and 44% to 55% reduction in lesion areas in non-sterile and sterile soils, respectively) (Figure 5C). Together, these data suggest that both *T. virens* and *P. chlororaphis* 30-84 colonization induced resistance to *C. graminicola* in both sterile and non-sterile soil, while insect defense was compromised only in the sterile soil condition.

### 2.5. T. virens and P. chlororaphis 30-84 Colonization Reduced Wound-Induced JA Only in Plants Grown in Sterile Soil

Decreased FAW resistance found with the amendment of *P. chlororaphis* 30-84 in this study prompted us to investigate whether this biocontrol agent impacts JA homeostasis in response to wounding in sterile soil. Phytohormone and oxylipin profiling revealed that *P. chlororaphis* 30-84 treatment reduced wound-induced accumulation of jasmonates, including 12-OPDA, JA, JA-Ile, and JA-Leu, in sterile soil (Figure 6A). More specifically, treatment with *T. virens*, *P. chlororaphis* 30-84, and the mixture of *T. virens* and *P. chlororaphis* 30-84 resulted in 25%, 21%, and 19% reductions in JA accumulation and 27%, 32%, and 27% reductions in JA-Ile level at 1 hpw, respectively (Figure 6B,C). Moreover, JA catabolism was increased by 50%, 13%, and 21% in the plants treated with *T. virens*, *P. chlororaphis* 30-84, and the mixture of *T. virens* and *P. chlororaphis* 30-84, respectively (Figure 6D). Interestingly, suppression of wound-induced JA biosynthesis (Figure 6) and FAW insect defense (Figure 5) by either *T. virens* or *P. chlororaphis* 30-84 was only observed in the sterile soil condition. A similar accumulation pattern was observed for 9,10-KODA (17% to 34% reduction) in sterile soil (Figure 6E). Together, these results suggest that *T. virens* and *P. chlororaphis* 30-84 suppress foliar-feeding insect defense via regulating JA homeostasis in response to wounding only in plants grown in sterile soil amended with either of these biocontrol agents.

### 2.6. T. virens Colonization Suppressed WCR Larval Survival and Weight Gain

Because *T. virens* colonizes roots endophytically and is known to produce a variety of secondary metabolites toxic to other organisms, we hypothesized that, unlike the effect of leaf herbivores, *T. virens* treatment may affect its interaction with a root-feeding insect, such as WCR larvae. The results showed that significantly fewer larvae were recovered (50% fewer) (Figure 7A), and WCR larvae gained significantly less weight after feeding on *T. virens*-colonized plants (57% less) (Figure 7B). However, *T. virens* treatment did not reduce tissue consumption (Figure 7C,D). These results suggest that *T. virens* may be considered as a potent biocontrol agent against WCR by suppressing WCR larval survival and weight gain.

## 3. Discussion

Soil-borne *Trichoderma* spp. are ubiquitous in soil worldwide, and have been used as biocontrol agents against pathogens for decades [23,34,35,36,37]. However, the potential utility of *Trichoderma* spp. as biocontrol agents against insect pests has just begun to be addressed [26], and information on their effects against foliar-feeding and root-feeding insects is scarce. Accumulating evidence suggests that root colonization by certain *Trichoderma* spp. triggers ISR against insects, such as aphids [38,39,40,41], whiteflies [42], thrips [43], and Lepidoptera [39,44,45,46,47,48], and also enhances defense against spider mites [49] and nematodes [50,51,52]. However, *T. virens* has not been tested for its effects on insects to date. Previous reports [27,28,30] and this study have shown that *T. virens* confers increased resistance to *C. graminicola*. In addition, these studies have identified a fungal-secreted peptide, Sm1, as a signal necessary for inducing systemic resistance by showing that the *Δsm1* deletion mutant of *T. virens* is unable to induce ISR [27,28,30]. In contrast to *Δsm1*, another signaling peptide mutant of *T. virens*, *Δsir1* [29,30], has been found to trigger enhanced ISR against *C. graminicola*. As this mutant overexpresses *SM1* transcripts [30], we hypothesized that the increased ISR following colonization by this mutant is due to elevated *SM1* expression. We tested this hypothesis by deleting *SM1* in the *Δsir1* mutant background and found that the *Δsm1Δsir1* double mutant still displayed stronger resistance as compared to the wild-type strain. We suggest that the Sir1 peptide acts to suppress ISR against *C. graminicola* not only by downregulating *SM1* expression, but also by employing other as-yet unknown mechanisms of ISR suppression.

Several reports have shown that several *Trichoderma* spp. directly impact Lepidoptera spp. by producing antifeedant compounds and toxic secondary metabolites [26,53,54,55,56,57], or indirectly impact them via activation of JA-mediated systemic defense, production of parasitoid-attracting volatile compounds, or disruption of insect gut microbiota and metabolomes [39,40,44,45,46,48,58]. Here, we demonstrated that treatment with *T. virens* suppressed defense against FAW by reducing wound-induced JA production, and increased JA catabolism in plants grown in sterile soil. Consistent with our findings, Lelio et al. (2021) [45] showed that when grown at cool temperatures (20 °C as compared to 25 °C), tomato plants colonized with *T. afroharzianum* T22 had less negative impact on *Spodoptera littoralis* larvae, and this was accompanied by suppressed JA biosynthesis and signaling pathways. Also consistent with our results, Kinyungu et al. (2023) [47] showed that treatment with the *T. atroviride* F2S21 strain resulted in a higher leaf defoliation rate by FAW larvae. The fact that some studies have shown increased resistance to certain chewing insects due to *Trichoderma* spp. treatment [39,44,48] while others, including the current study, have shown an opposite effect of *Trichoderma* treatment, suggests that the effects of *Trichoderma* on plant interactions with insect herbivores are species- and strain-specific, as well as environment-dependent. Interestingly, our results indicate that the observed reduced insect defense is independent of the fungal secreted peptides Sm1 and Sir1, suggesting that fungal signals regulating ISR against pathogens and induced susceptibility against chewing insects involve different molecular signals, to be identified in the future.

Previous reports have shown that root colonization by *T. virens* induces biosynthesis of the JA precursor, 12-OPDA, and ketols [30,59]. Both 12-OPDA and ketols have been shown to act as xylem-mobile long-distance signals that prime the above-ground organs for greater resistance against pathogens in maize [30,59]. In contrast to these oxylipins, JA-Ile levels in xylem sap were not significantly increased in response to *T. virens* treatment, and transfusion of maize seedlings with xylem sap supplemented with JA-Ile increased their susceptibility to *C. graminicola* [30]. The role of JA as a susceptibility factor against anthracnose leaf blight caused by *C. graminicola* was further supported by the studies of Gorman et al. (2020) [12] and Huang et al. (2023) [13]. Moreover, transcriptome analysis has shown that maize roots respond to colonization by *T. virens* with increased expression of 12-OPDA biosynthesis genes, but suppressed expression of the genes required for 12-OPDA conversion to JA-Ile [30]. Because JA is widely reported to suppress growth in numerous plant species [60,61,62], it is tempting to speculate that attenuated synthesis of JAs in roots colonized by *T. virens* is a potential mechanism accounting for the growth promotion effect conferred by this symbiont, as observed in this study and previous reports. Similar to the *T. virens*-mediated induction of growth promotion and ISR against pathogens, we showed that root colonization by the bacterial symbiont *P. chlororaphis* 30-84 resulted in increased seedling growth and systemic resistance against *C. graminicola*, suggesting a similar mechanism underlying these two beneficial responses. However, as in the case of *T. virens*, *P. chlororaphis* 30-84 treatment reduced JA production and decreased resistance to herbivory by FAW when treated plants were grown in sterile soil. Therefore, it is likely that the induction of growth promotion and ISR against the hemibiotrophic pathogen *C. graminicola* by both of the beneficial microorganisms, *T. virens* and *P. chlororaphis* 30-84, is linked to their ability to suppress JA synthesis and JA-mediated signaling. Interestingly, wound-induced JA accumulation and insect defense were not affected when maize seedlings were treated with *T. virens* or *P. chlororaphis* 30-84 in non-sterile soil, suggesting that natural microbial communities dampened the JA suppression activities of both endophytes.

WCR root feeding causes substantial economic damage to maize production in the United States and Europe [63,64], and the insect has adapted to overcome most of the available management techniques, including chemical insecticides and *Bacillus thuringiensis* (Bt) toxins [64,65]. So far, there has been a lack of efficient management of this notorious pest and, thus, there is an urgent need for a sustainable and economical management approach, such as utilizing efficient biocontrol agents. In this study, we showed that *T. virens* colonization significantly enhanced the mortality of, and suppressed weight gain in, WCR larvae. Our previous report showed that increased WCR mortality was correlated with greater levels of herbivory-induced production of insecticidal death acids and ketols [16]. Therefore, one of the potential mechanisms of *T. virens*-mediated WCR defense may be activation of the production of insecticidal defensive compounds. The other potential mechanism behind the detrimental effect on larvae survival and growth conferred by *T. virens* may be the production of some fungus-derived insecticidal metabolites, as reported in several studies [26,53,54,55,56,57]. Further experiments are needed to explore whether *T. virens* has a direct insecticidal impact on the WCR root-feeding larvae, or has an indirect effect due to the induction of the host-mediated synthesis of toxic compounds.

## 4. Materials and Methods

### 4.1. Plant, Growth Medium, and Plant-Beneficial Inoculants

The maize (*Zea mays*) inbred line B73 and *opr7opr8* double mutant (Yan et al., 2012) [14] were used in this study. Plants were grown in conical pots (20.5 × 4 cm) filled with autoclaved (121 °C for 60 min, two cycles) commercial potting mix (Jolly Gardener Pro Line C/20) on light shelves at room temperature (22–24 °C) with a 16 h light period. To test the efficacy of the beneficial microorganisms in an agricultural setting, the maize B73 seeds were grown in a mixture of field soil (collected from Texas A&M University Research Farm in the fall of 2022 after maize was harvested) and potting mix (Jolly Gardener Pro Line C/20) at a ratio of 1 to 3. The mixture was either autoclaved (121 °C for 60 min, two cycles; henceforth referred to as sterile) or untreated (hence referred to as non-sterile) after mixing. Chlamydospores of *T. virens* strains TvWT (Gv29-8; Baek and Kenerly, 1998) [66], *Δsm1* (Djonovic’ et al., 2006; Djonovic et al., 2007) [27,28], *Δsir1* (formerly D77560; Lamdan et al., 2015) [29], and *Δsm1Δsir1* (this study) were obtained from liquid cultures grown in molasses-yeast extract medium as described in Wang et al. (2020) [30]. *P. chlororaphis* 30-84 was grown in Luria-Bertani (LB) medium containing 5 g of NaCl per liter at pH 7 and 28 °C with agitation at 200 rpm, as described in Yuan et al. (2020) [33]. Seven days after sowing, seedlings were treated with 0.05 g of *T. virens* chlamydospores, or 300 μL of *P. chlororaphis* 30-84 suspension at OD_620_ = 0.8, or a mixture of both *T. virens* and *P. chlororaphis* in the same amounts as described above (added to the soil at a depth of approximately 2 cm around the seeds), or left untreated.

### 4.2. Anthracnose Leaf Blight Assay

*C. graminicola* strain 1.001 was grown on half-strength potato dextrose agar (PDA) plates for 2–3 weeks to allow sporulation. Two weeks after being colonized by beneficial microorganisms, the plants were inoculated with 10 μL of conidial suspension (10^6^ conidia/mL) with six droplets of spore suspensions per leaf collected from hemibiotrophic fungal pathogen *C. graminicola* plates, as described in Huang et al. (2023a) [13]. The leaves were scanned and the lesion areas were measured using ImageJ software (IJ 1.46r) [67] seven days post-inoculation.

### 4.3. Fall Armyworm Assay

The laboratory strain of fall armyworm (FAW; *Spodoptera frugiperda*) was purchased from Benzon Research (Carlisle, PA, USA). Two weeks after incubation with the beneficial microorganisms, six FAW neonates were placed to the whorls of maize seedlings contained in individual plastic jars and allowed to move and feed on the plants freely for 7 days, as described in Huang et al. (2023b) [16]. Seven days post-infestation, FAW larvae were removed from the plants, and their total weights were determined. To evaluate insect resistance and leaf damaged area, one maize leaf was individually caged in a handmade clip-cage and infested with one third-instar FAW larva per spot for approximately one hour, then moved toward the base of the plant. Leaves were then scanned, and eaten areas were measured with ImageJ software [67].

### 4.4. Oxylipin Profiling of Wounded Leaf Tissue

For the wounding treatment, the third fully expanded leaf of seedlings grown in sterile or non-sterile soil conditions that were previously colonized with beneficial microorganisms for two weeks or left untreated was wounded seven times using a hemostat, with three wound sites on one side and four on the other side of the midvein with the wound sites approximately 1 cm apart. The wounded regions were then harvested and placed in 2 mL screw-cap Fast-Prep tubes (Qbiogene, Carlsbad, CA, USA) in liquid nitrogen and stored in a −80 °C freezer. Phytohormone and oxylipin extraction and profiling of wounded leaf tissue were performed as described previously [13].

### 4.5. Western Corn Rootworm Bioassays

Western corn rootworm (WCR; *Diabrotica virgifera virgifera*) eggs were provided by USDA-ARS North Central Agriculture Research Laboratory (Brookings, SD, USA). Two weeks after being colonized by beneficial microorganisms, ten WCR neonates were placed on the soil surface in individual cone-tainers with a maize seedling and allowed to burrow and feed for ten days. WCR larvae were removed from the soil, and the total weight was determined ten days post-infestation. The fresh weights of shoot and root tissues were measured after the soil was remove, and the ratio was analyzed as larvae damaged over control root mass, as described in Huang et al. (2023) [16].

### 4.6. Statistical Methods

Statistical analyses were performed using the software program JMP Pro 17 (SAS Institute Inc., Cary, NC, USA). Lesion areas, FAW larval weight, leaf eaten areas, wound-induced accumulation of metabolites, and plant heights were analyzed using one-way analysis of variance (ANOVA) with post-hoc Tukey HSD (honestly significant difference). FAW larval weight after feeding on *opr7opr8* double mutants and wild-type seedlings treated with *T. virens* or left untreated, along with root and shoot ratios after WCR infestation, were analyzed using two-way ANOVA with Tukey HSD. The effect of sterile and non-sterile mixture was analyzed using a Student’s *t*-test.

## 5. Conclusions

The elevated level of ISR against *C. graminicola* triggered by both *T. virens Δsir1* and *Δsm1Δsir1* double mutants revealed that a functional Sir1 signal peptide reduces disease resistance by not only downregulating *SM1* (as demonstrated previously by Wang et al. (2020)), but via other unknown mechanisms as well. The colonization of plants grown in sterile soil by different *T. virens* mutants suppressed wound-induced JA accumulation, resulting in reduced defense against FAW, which was independent of both signal peptides, Sm1 and Sir1. When testing both of the beneficial microorganisms *T. virens* and *P. chlororaphis* 30-84 using both sterile and non-sterile mixtures of field soil and potting mix, both biocontrol agents triggered ISR against *C. graminicola* in both soil types. However, these agents reduced defense against FAW only when the treated plants were grown in sterile soil due to reduced JA. In contrast to the effect on leaf-feeding FAW, *T. virens* colonization suppressed root-feeding WCR larvae survival and weight gain. This marks the first report revealing the potential of *T. virens* as a biocontrol agent against WCR.

## Figures and Tables

**Figure 1 plants-13-01240-f001:**
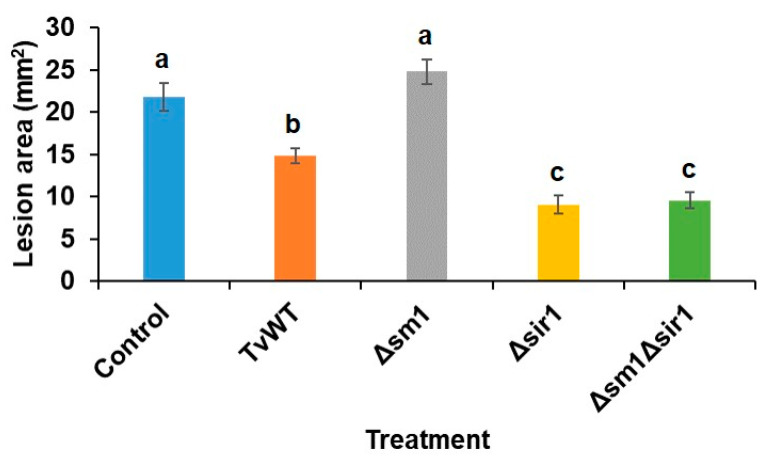
Both *T. virens Δsir1* and *Δsm1Δsir1* treatments elicited a greater level of ISR against *C. graminicola* than TvWT. Lesion areas of maize seedlings colonized by different *T. virens* strains after inoculation with *C. graminicola* for 7 days. Bars are means ± SE (*n* ≥ 23). Different letters indicate statistically significant differences (Tukey’s HSD test, *p* < 0.05).

**Figure 2 plants-13-01240-f002:**
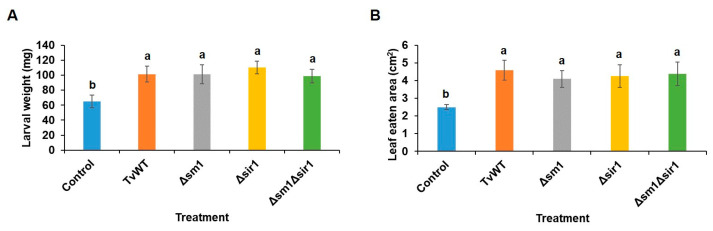
*T. virens* colonization suppressed maize defense against insect herbivory. One-week old maize seedlings were colonized by *T. virens* strains for 14 days before FAW infestation. (**A**) FAW larval weight was determined after neonates feeding on maize plants for 7 days (*n* ≥ 23). (**B**) Maize leaf consumed area by 3rd-instar FAW larvae for 6 h (*n* ≥ 4). Bars are means ± SE. Different letters indicate statistically significant differences (Tukey’s HSD test, *p* < 0.05).

**Figure 3 plants-13-01240-f003:**
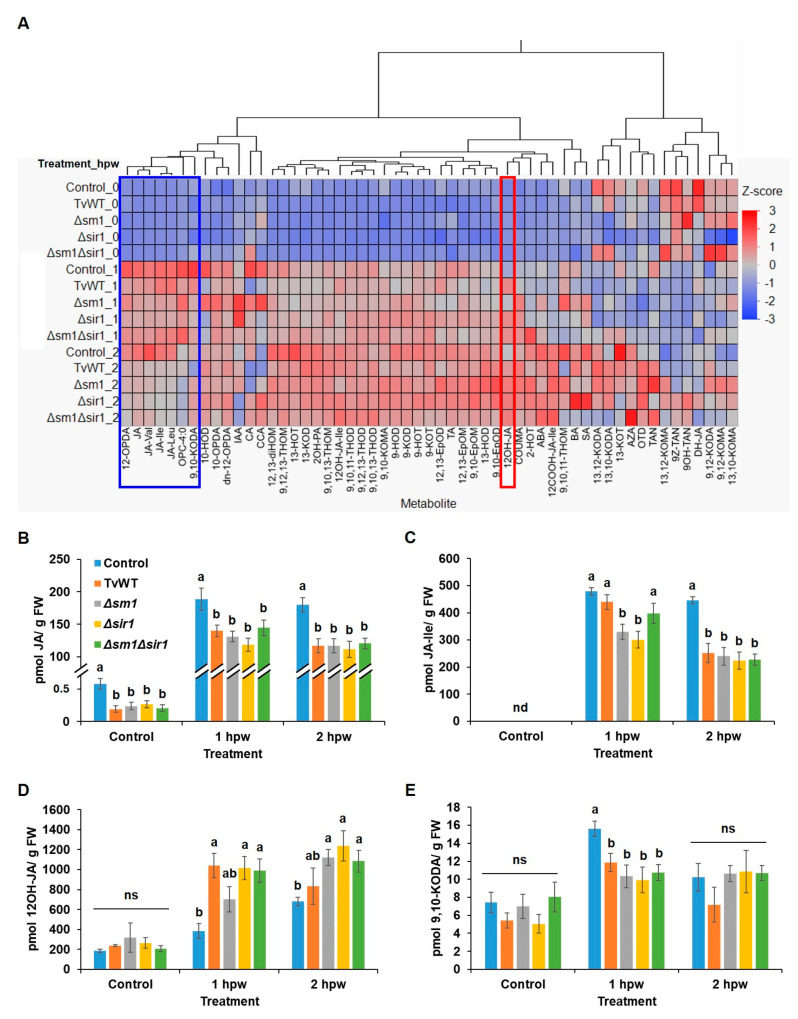
*T. virens* colonization suppressed wound-induced 9,10-KODA and JA biosynthesis and enhanced JA catabolism. (**A**) Heatmap showing relative accumulation of oxylipins and phytohormones in leaves of B73 seedlings colonized by different *T. virens* stains or untreated control at 0, 1, and 2 h post wounding. Contents of (**B**) JA, (**C**) JA-Ile, (**D**) 12OH-JA, and (**E**) 9,10-KODA in different *T. virens* strain-colonized or control plants at 0, 1, and 2 h post wounding. Abbreviations for compounds are listed in Appendix A. Bars are means ± SE (*n* = 6). Different letters indicate statistically significant differences within the same time points (Tukey’s HSD test, *p* < 0.05). Abbreviations for statistical analysis: ns, not significant; nd, signal not detected.

**Figure 4 plants-13-01240-f004:**
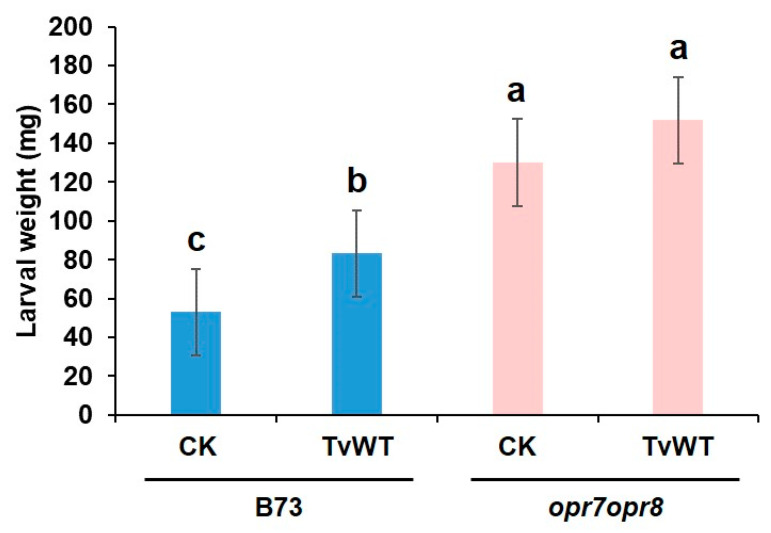
Reduced resistance against FAW in response to *T. virens* colonization is modulated by JA signaling. Maize B73 wild-type or *opr7opr8* double mutant seedlings were colonized by TvWT for 14 days or left as untreated control (CK) before FAW infestation. FAW larval weight was determined after neonates feeding on maize plants for 7 days. Bars are means ± SE (*n* ≥ 13). Different letters indicate statistically significant differences on log-transformed data (two-way ANOVA with Tukey’s HSD test, *p* < 0.05).

**Figure 5 plants-13-01240-f005:**
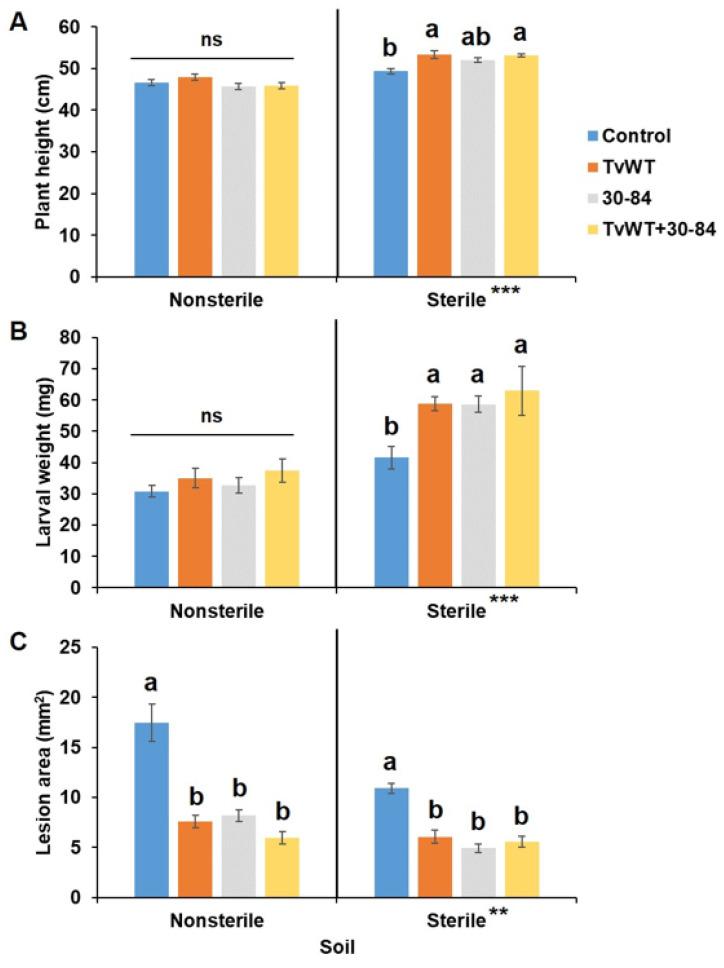
*T. virens* and *P. chlororaphis* 30-84 colonization enhanced growth but suppressed defense against FAW only in sterile soil, and increased resistance to *C. graminicola* in both soil types. Maize seedlings were colonized by TvWT, *P chlororaphis* 30-84, or both for 14 days before plant height measurement and *C. graminicola* infection or FAW infestation. (**A**) Plant height measured at 14 days after colonization (*n* ≥ 8). (**B**) FAW larval weight (*n* ≥ 12) and (**C**) lesion areas (*n* = 18) were determined at 7 days after infection or infestation. Bars are means ± SE. Different letters indicate statistically significant differences on log-transformed data within the same treatment (Tukey’s HSD test, *p* < 0.05). Asterisks represent statistically significant differences between sterile and non-sterile soils (Student’s *t*-test, ** *p* < 0.01, *** *p* < 0.001). Abbreviations for statistical analysis: ns, not significant.

**Figure 6 plants-13-01240-f006:**
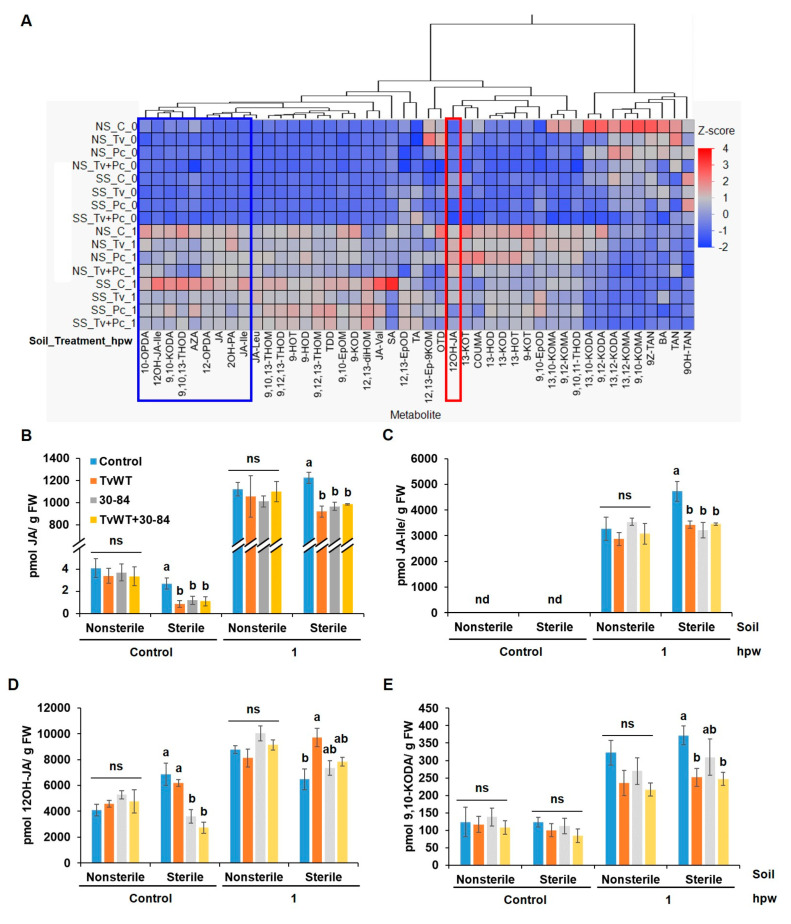
*T. virens* and *P. chlororaphis* 30-84 colonization suppressed wound-induced JA biosynthesis and enhanced JA catabolism in sterile soil. (**A**) Heatmap showing relative accumulation of oxylipins and phytohormones. Contents of (**B**) JA, (**C**) JA-Ile, (**D**) 12OH-JA, and (**E**) 9,10-KODA in plants colonized by *T. virens*, *P. chlororaphis* 30-84, and both or control in response to wounding at 0 and 1 hpw. Abbreviations for samples: NS, non-sterile soil; SS, sterile soil. Abbreviations for compounds are listed in Appendix A. Bars are means ± SE (*n* = 4). Different letters indicate statistically significant differences within the same treatment per time point (Tukey’s HSD test, *p* < 0.05). Abbreviations for statistical analysis: ns, not significant; nd, signal not detected.

**Figure 7 plants-13-01240-f007:**
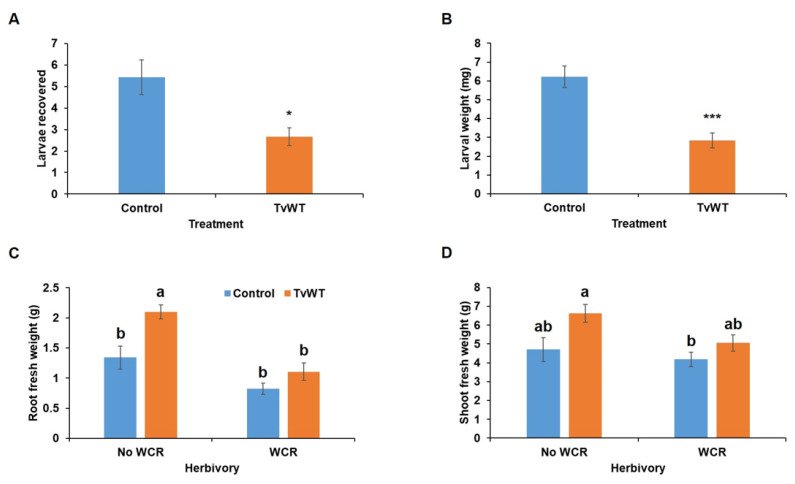
*T. virens* colonization suppressed WCR larval survival and weight gain. (**A**) Larvae recovery (*n* = 7), (**B**) larval mass (*n* = 7), (**C**) root (*n* ≥ 4), and (**D**) shoot ratios (*n* ≥ 4) were recorded 10 days post-infestation. Bars are means ± SE. Asterisks represent statistically significant differences (Student’s *t*-test, * *p* < 0.05, *** *p* < 0.001). Different letters indicate statistically significant differences (two-way ANOVA with Tukey’s HSD test, *p* < 0.05).

## Data Availability

The datasets used and analyzed during the current study are available from the corresponding author upon reasonable request.

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
