# Peer review of "Trichoderma virens and Pseudomonas chlororaphis Differentially Regulate Maize Resistance to Anthracnose Leaf Blight and Insect Herbivores When Grown in Sterile versus Non-Sterile Soils"

_plants, 2024, doi:10.3390/plants13091240_

Round 1
Reviewer 1 Report
Comments and Suggestions for Authors
The manuscript with the title: “Trichoderma virens and Pseudomonas chlororaphis differentially regulate maize resistance to anthracnose leaf blight and insect herbivores when grown in sterile versus non-sterile soils” is an interesting study, whose main innovation is that the effect of two biocontrol agents is observed under two growth conditions when maize is grown in sterile versus non-sterile soils. A series of experiments were conducted to observe the effect of each biological control agent, their effect on insects and fungi, and the quantity of that effect in terms of soil sterilization. The high quality of the work also lies in the use of numerous mutants, which allows the discovery of molecular mechanisms that lie in the dreams of the interactions studied. A major shortcoming is the way in which the results are presented. Numerous corrections need to be made here and the text needs to be refined in many places (this is discussed in the following text). Therefore, I believe that the text is not ready for publication and my decision is that the paper should undergo a major revision.
Major concerns:
Abstract – The abstract needs to be changed in detail to be clearer and more informative. Introduction – The introduction should clearly state the aims of the study without presenting and discussing the results obtained. In the last paragraph of the introduction, the authors write: “The main aim of this study was to investigate the effects of T. virens and its secreted peptide signaling on the production of wound-induced JA and other defensive oxylipin metabolites and to test whether the beneficial effects of this endophyte also increase resistance to FAW and WCR.” Then the authors go on to write about some results obtained and conclusions drawn regarding the endophyte strain Pseudomonas chlororaphis 30-84 and another fungal pathogen Colletotrichum graminicola. In addition, we have sterile and non-sterile conditions. So what was the main objective of the study that readers could not find out? This brings further disadvantages because if it is not clear what the aim of the work is, what is being done and why, then the presentation of the results themselves is unclear and confusing.
Results – I think that the results should be presented with many changes. First, the text starts with a few sentences of discussion in each subheading; this is not the place for it. Second, the presentation of the results is confusing, unclear, and hard to follow. Although the title of the paper states that activity against FAW and WCR was studied, the first section in the results refers to activity against the fungal pathogen Colletotrichum graminicola. Why? How does this result and the work on this pathogen relate to the title and the protection of plants against FAW and WCR? It is not clear from anything that has been written so far. I thought similar suggestions could be made in all the results sections. So please remove the discussion from the results and avoid repetition, discussion and numerous abbreviations when presenting the results. Also, some of the results point in a different direction than the reader might expect from the title. Does treatment with Trichoderma virens and Pseudomonas chlororaphis protect plants from FAW and/or WCR or not? It was hard to be sure what the right answers was.
Discussion and Conclusion – When authors made clearly idea about what was the aim(s) in this paper than they should write this two section accordingly. Otherwise at this moment they are both as separate sections.
Author Response
Reviewer 1
Comments and Suggestions for Authors:
The manuscript with the title: “Trichoderma virens and Pseudomonas chlororaphis differentially regulate maize resistance to anthracnose leaf blight and insect herbivores when grown in sterile versus non-sterile soils” is an interesting study, whose main innovation is that the effect of two biocontrol agents is observed under two growth conditions when maize is grown in sterile versus non-sterile soils. A series of experiments were conducted to observe the effect of each biological control agent, their effect on insects and fungi, and the quantity of that effect in terms of soil sterilization. The high quality of the work also lies in the use of numerous mutants, which allows the discovery of molecular mechanisms that lie in the dreams of the interactions studied. A major shortcoming is the way in which the results are presented. Numerous corrections need to be made here and the text needs to be refined in many places (this is discussed in the following text). Therefore, I believe that the text is not ready for publication and my decision is that the paper should undergo a major revision.
We appreciated the time and effort of the reviewer in reviewing this manuscript. Your suggestions helped us improve this manuscript. We have clarified the objectives in the sequence of the study as they emerged based on the results obtained and we presented those objectives in a logical manner in the last paragraph of the Introduction and wherever it was relevant in Results.
Here is the rational reasoning behind specific objectives to address the valid concern of the reviewer. It is indeed important to emphasize the logic behind those objectives and how the new objectives have been put forward based on the results obtained.
Previous studies showed that two T. virens secreted peptide signals, Sm1 and Sir1, play important roles in regulating induced systemic resistance against pathogens and their interactions are not fully understood and whether they contribute to defense against FAW remained unexplored. Therefore, the initial objective was to investigate the effects of T. virens and its secreted peptide signals on the wound-induced JA biosynthesis and insect defense against FAW.
First, we tested the hypothesis whether enhanced ISR by Δsir1 is due to elevated SM1 expression in this strain.
Second, the follow-up experiment was carried out to test whether colonization of these different T. virens strains trigger differential defense responses against FAW. Surprisingly, colonization of these T. virens strains reduced FAW defense, regardless of the mutation of these two signaling peptides. Oxylipin and phytohormone profiling revealed that the reduced defense against FAW was associated with suppressed wound-induced JA production of the T. virens colonized plants.
Third, because we did not expect to see negative impact on insect resistance, we further tested whether other growth promoting microbes have a similar effect. For this, we used a bacterial biocontrol agent, P. chlororaphis, and expanded the experimental design to test the impact of these two agents in sterile and non-sterile soil mixtures. The results showed that colonization of either T. virens or P. chlororaphis 30-84 suppressed wound-induced JA production and defense against FAW only when plants were grown in sterile soil while no significant impact was observed in non-sterile soil.
Fourth, because soil sterility altered interactions with FAW, we tested the impact of these two biocontrol agents in triggering ISR to fungal pathogen, C. graminicola in both sterile and non-sterile soil conditions. The results showed that colonization of T. virens and P. chlororaphis 30-84 induced ISR to C. graminicola in both soil conditions.
Last, we showed that T. virens suppressed WCR larval survival and weight gain, suggesting the potential role of T. virens as a biocontrol agent against WCR.
Major concerns:
Abstract – The abstract needs to be changed in detail to be clearer and more informative.
Introduction – The introduction should clearly state the aims of the study without presenting and discussing the results obtained. In the last paragraph of the introduction, the authors write: “The main aim of this study was to investigate the effects of T. virens and its secreted peptide signaling on the production of wound-induced JA and other defensive oxylipin metabolites and to test whether the beneficial effects of this endophyte also increase resistance to FAW and WCR.” Then the authors go on to write about some results obtained and conclusions drawn regarding the endophyte strain Pseudomonas chlororaphis 30-84 and another fungal pathogen Colletotrichum graminicola. In addition, we have sterile and non-sterile conditions. So what was the main objective of the study that readers could not find out? This brings further disadvantages because if it is not clear what the aim of the work is, what is being done and why, then the presentation of the results themselves is unclear and confusing.
Results – I think that the results should be presented with many changes. First, the text starts with a few sentences of discussion in each subheading; this is not the place for it. Second, the presentation of the results is confusing, unclear, and hard to follow. Although the title of the paper states that activity against FAW and WCR was studied, the first section in the results refers to activity against the fungal pathogen Colletotrichum graminicola. Why? How does this result and the work on this pathogen relate to the title and the protection of plants against FAW and WCR? It is not clear from anything that has been written so far. I thought similar suggestions could be made in all the results sections. So please remove the discussion from the results and avoid repetition, discussion and numerous abbreviations when presenting the results. Also, some of the results point in a different direction than the reader might expect from the title. Does treatment with Trichoderma virens and Pseudomonas chlororaphis protect plants from FAW and/or WCR or not? It was hard to be sure what the right answers was.
Discussion and Conclusion – When authors made clearly idea about what was the aim(s) in this paper than they should write this two section accordingly. Otherwise at this moment they are both as separate sections.
We have revised the entire manuscript substantially according to your valuable suggestion and the edits are track changes.
Reviewer 2 Report
Comments and Suggestions for Authors
This research characterizes the role of Trichoderma virens and Pseudomonas chlororaphis in maize resistance to anthracnose leaf blight and insect herbivores. With changes in climatic conditions and constantly evolving plant pathogens, there is the need for more biocontrol measures for crop protection. The findings of this research add more knowledge about the potential of these biocontrol fungal and bacterial strains.
There are some corrections to be taken care of in the manuscript.
Be consistent with how you mention maize in the manuscript. Either mention as maize or corn, and not both.
Lines 62-63: “However, using biocontrol agents against pests has only recently been considered…” is not correct. Author’s have to be careful while writing such statements. Be specific that the biocontrol effect of Trichoderma on insect resistance remains largely unexplored.
Figures: Mention how many replicates were tested, such as n = ?, in the figures where applicable. Maintain font size and style in all the figures. The statistical significance mentioned should also be consistent. Either keep them as alphabet separation or as ***.
Line 188, 351: stain to correct as strain (s)
Including figures of leaf lesions, larval weight if they were visually different, as observed in the various treatments would be useful for the readers as a point of reference.
Comments on the Quality of English LanguageMinor spell checks
Author Response
Reviewer 2
Comments and Suggestions for Authors
This research characterizes the role of Trichoderma virens and Pseudomonas chlororaphis in maize resistance to anthracnose leaf blight and insect herbivores. With changes in climatic conditions and constantly evolving plant pathogens, there is the need for more biocontrol measures for crop protection. The findings of this research add more knowledge about the potential of these biocontrol fungal and bacterial strains.
We would like to thank the reviewer for taking the necessary time and effort to review the manuscript.
There are some corrections to be taken care of in the manuscript.
Be consistent with how you mention maize in the manuscript. Either mention as maize or corn, and not both.
We have changed corn into maize throughout the manuscript which is a preferred term. However, western corn rootworm which is the common name of the pest and corn used in the references stayed unchanged.
Lines 62-63: “However, using biocontrol agents against pests has only recently been considered…” is not correct. Author’s have to be careful while writing such statements. Be specific that the biocontrol effect of Trichoderma on insect resistance remains largely unexplored.
The sentence has been revised according to the suggestion:
“However, using Trichoderma spp. as biocontrol agents against pests has only recently been considered, and their impact on defense against chewing insects remains largely unexplored [25,26].”
Figures: Mention how many replicates were tested, such as n = ?, in the figures where applicable. Maintain font size and style in all the figures. The statistical significance mentioned should also be consistent. Either keep them as alphabet separation or as ***.
Number of replicates of each experiment has been added to the figure legends. However, the statistical significance using alphabet separation or asterisks stay unchanged because they were generated using either Tukey’s HSD test to show the difference among different treatments using alphabets or student’s t-test to show any difference in pair-wise comparison.
Line 188, 351: stain to correct as strain (s)
Corrected.
Including figures of leaf lesions, larval weight if they were visually different, as observed in the various treatments would be useful for the readers as a point of reference.
Thank you for bringing this point to attention. Unfortunately, FAW larvae were not preserved after fresh weight measurement, we are not able to provide the images showing the contrast of the larval size among the treatments. Therefore, we decided to present only the figures without the images.
Reviewer 3 Report
Comments and Suggestions for Authors
The manuscript is in very good condition.
Lines 35-36: scientific names of organisms should be italicized.
Line 351: strain, not stain
Line 426: the references should be modified to fit the style of the journal. There are many scientific names of organisms that are not italicized. In addition, titles of some references are capitalized.
Line 569: the journal name and page numbers are missing.
Author Response
Reviewer 3
The manuscript is in very good condition.
We would like to thank the reviewer for taking the necessary time and effort to review the manuscript and we have corrected the manuscript according to your suggestions.
Lines 35-36: scientific names of organisms should be italicized.
Line 351: strain, not stain
Line 426: the references should be modified to fit the style of the journal. There are many scientific names of organisms that are not italicized. In addition, titles of some references are capitalized.
Line 569: the journal name and page numbers are missing.
Round 2
Reviewer 1 Report
Comments and Suggestions for Authors After a thorough review of the revised version of the paper, I believe that the paper has not been corrected in all the places where this was suggested, and I think that this is necessary for the paper to be ready for publication.Author Response
Reviewer 1
Round 2
After a thorough review of the revised version of the paper, I believe that the paper has not been corrected in all the places where this was suggested, and I think that this is necessary for the paper to be ready for publication.
We have provided point-to-point responses to your specific suggestions and concerns in previous review report.
Round 1
Comments and Suggestions for Authors:
The manuscript with the title: “Trichoderma virens and Pseudomonas chlororaphis differentially regulate maize resistance to anthracnose leaf blight and insect herbivores when grown in sterile versus non-sterile soils” is an interesting study, whose main innovation is that the effect of two biocontrol agents is observed under two growth conditions when maize is grown in sterile versus non-sterile soils. A series of experiments were conducted to observe the effect of each biological control agent, their effect on insects and fungi, and the quantity of that effect in terms of soil sterilization. The high quality of the work also lies in the use of numerous mutants, which allows the discovery of molecular mechanisms that lie in the dreams of the interactions studied. A major shortcoming is the way in which the results are presented. Numerous corrections need to be made here and the text needs to be refined in many places (this is discussed in the following text). Therefore, I believe that the text is not ready for publication and my decision is that the paper should undergo a major revision.
We appreciated the time and effort of the reviewer in reviewing this manuscript. Your suggestions helped us improve this manuscript. We have clarified the objectives in the sequence of the study as they emerged based on the results obtained and we presented those objectives in a logical manner in the last paragraph of the Introduction and wherever it was relevant in Results.
Here is the rational reasoning behind specific objectives to address the valid concern of the reviewer. It is indeed important to emphasize the logic behind those objectives and how the new objectives have been put forward based on the results obtained.
Previous studies showed that two T. virens secreted peptide signals, Sm1 and Sir1, play important roles in regulating induced systemic resistance against pathogens and their interactions are not fully understood and whether they contribute to defense against FAW remained unexplored. Therefore, the initial objective was to investigate the effects of T. virens and its secreted peptide signals on the wound-induced JA biosynthesis and insect defense against FAW.
First, we tested the hypothesis whether enhanced ISR by Δsir1 is due to elevated SM1 expression in this strain.
Second, the follow-up experiment was carried out to test whether colonization of these different T. virens strains trigger differential defense responses against FAW. Surprisingly, colonization of these T. virens strains reduced FAW defense, regardless of the mutation of these two signaling peptides. Oxylipin and phytohormone profiling revealed that the reduced defense against FAW was associated with suppressed wound-induced JA production of the T. virens colonized plants.
Third, because we did not expect to see negative impact on insect resistance, we further tested whether other growth promoting microbes have a similar effect. For this, we used a bacterial biocontrol agent, P. chlororaphis, and expanded the experimental design to test the impact of these two agents in sterile and non-sterile soil mixtures. The results showed that colonization of either T. virens or P. chlororaphis 30-84 suppressed wound-induced JA production and defense against FAW only when plants were grown in sterile soil while no significant impact was observed in non-sterile soil.
Fourth, because soil sterility altered interactions with FAW, we tested the impact of these two biocontrol agents in triggering ISR to fungal pathogen, C. graminicola in both sterile and non-sterile soil conditions. The results showed that colonization of T. virens and P. chlororaphis 30-84 induced ISR to C. graminicola in both soil conditions.
Last, we showed that T. virens suppressed WCR larval survival and weight gain, suggesting the potential role of T. virens as a biocontrol agent against WCR.
We have revised the entire manuscript substantially according to your valuable suggestion and the edits are track changes and listed in each section below.
Major concerns:
Abstract – The abstract needs to be changed in detail to be clearer and more informative.
Line 29-33: “Further comparative analyses of the systemic effects of these endophytes when applied into sterile or non-sterile field soil showed that both T. virens and P. chlororaphis 30-84 triggered ISR against hemibiotrophic fungal pathogen C. graminicola in both soil conditions, but only suppressed JA production and resistance to FAW in sterile soil while no significant impact was observed when applied in non-sterile soil.”
Introduction – The introduction should clearly state the aims of the study without presenting and discussing the results obtained. In the last paragraph of the introduction, the authors write: “The main aim of this study was to investigate the effects of T. virens and its secreted peptide signaling on the production of wound-induced JA and other defensive oxylipin metabolites and to test whether the beneficial effects of this endophyte also increase resistance to FAW and WCR.” Then the authors go on to write about some results obtained and conclusions drawn regarding the endophyte strain Pseudomonas chlororaphis 30-84 and another fungal pathogen Colletotrichum graminicola. In addition, we have sterile and non-sterile conditions. So what was the main objective of the study that readers could not find out? This brings further disadvantages because if it is not clear what the aim of the work is, what is being done and why, then the presentation of the results themselves is unclear and confusing.
We have revised the last paragraph of the introduction according to the reviewer’s suggestion as shown below.
Line 80-99:
The original objective of this study was to investigate the effects of T. virens and its secreted peptide signals on the production of wound-induced JA and other defensive oxylipin metabolites and to test whether the beneficial effects of this endophyte extend to increasing resistance against FAW. Contrary to our expectation, the results demonstrated that maize seedlings grown in sterile soil amended with wild-type T. virens (TvWT), the sm1 or sir1 single or the double mutants were found to suppress wound-induced JA biosynthesis, resulting in reduced resistance to FAW. Such unexpected results prompted us to further test whether such a detrimental effect on defense against herbivory could be ascribed to another well-studied growth-promoting bacterial endophyte, P. chlororaphis. Colonization of roots by the P. chlororaphis 30-84 also reduced production of JA and defense against FAW in sterile soil. These results necessitated the next objective, which was to test whether reduced defense against herbivory can also be observed in plants grown under non-sterile soil conditions. The results showed no detrimental change in resistance to FAW. Such differential impact of soil sterility prompted us to test whether soil conditions alter the effectiveness of ISR triggered by both biocontrol agents against the leaf pathogen, C. graminicola, and the results showed strong induction of ISR regardless of soil condition. The final objective of this study was to test whether T. virens treatment impacts maize interactions with the root-feeding WCR larvae. We showed that colonization of roots by T. virens reduced larvae survival and growth.
Results – I think that the results should be presented with many changes. First, the text starts with a few sentences of discussion in each subheading; this is not the place for it.
We have rewritten introduction sentences for each of the Result sections to provide the unambiguous rationale and the purpose for each experiment. As suggested, we deleted repeated literature review statements in all Result sections. As to the Title, we entertained an idea of a more conclusion-like Title but could come up with a title that would succinctly and accurately describe the diverse results, so we left the Title as is.
Specifically, in the Results section, we made the following changes primarily to explain why we ran specific experiments:
2.1.
Line 103-106: repetition of the introduction of these few sentences has been deleted.
2.2.
Line 123-125: A new introductory sentence reads as follows: “Because the effect of T. virens on ISR against insect herbivores has not been explored before, we tested whether TvWT and the secreted peptide signaling mutants Δsm1, Δsir1, and Δsm1Δsir1 induce differential resistance to the foliar-feeding FAW”.
2.3.
Line139-141: A new introductory sentence reads as follows: “Because T. virens treatment decreased insect defense and JA dominantly modulates defense responses against chewing insects [14], we tested whether T. virens colonization regulates wound-induced JA production in leaves”.
2.4.
Line 183-186: Subtitle has been revised as
“T. virens and P. chlororaphis colonization induced resistance to C. graminicola in both sterile and non-sterile soil conditions but enhanced growth and suppressed insect defense against FAW only in sterile soil”
Line 187-192:
A new introductory sentence reads as follows: ”Because of the detrimental effect of the growth promoting T. virens on herbivory resistance, we next tested whether another growth-promoting microorganism would have a similar impact on defense. For this, we chose the bacterial endophyte P. chlororaphis 30-84 that was reported to promote plant growth and suppress growth of fungal pathogens [32, 33], yet its role in insect defense against foliar-feeding insects remains mostly unexplored.”
Line 205-208:
“Together, these data suggested that both T. virens and P. chlororaphis 30-84 colonization induced resistance to C. graminicola in both sterile and non-sterile soil while insect defense was compromised only in sterile soil condition.”
2.5.
Line 219-220: Subtitle has been revised as
“T. virens and P. chlororaphis 30-84 colonization reduced wound-induced JA only in plants grown in sterile soil”
2.6.
Line 248-251:
A new introductory sentence reads as follows: “Because T. virens colonizes roots endophytically and is known to produce a variety of secondary metabolites toxic to other organisms, we hypothesized that, unlike the effect of leaf herbivores, T. virens treatment may affect its interaction with a root-feeding insect such as WCR larvae.”
Second, the presentation of the results is confusing, unclear, and hard to follow. Although the title of the paper states that activity against FAW and WCR was studied, the first section in the results refers to activity against the fungal pathogen Colletotrichum graminicola. Why?
Because our previous research on characterizing the functions of the secreted peptides, Sm1 and Sir1, were performed in terms of their differential regulation of ISR against Colletotrichum graminicola, we first investigated the hypothesized antagonistic interaction of these two peptides by creating Δsm1Δsir1 double mutant and tested its impact on ISR against this pathogen before this new mutant can be tested for relevance to insect defense. We could present it as a Supplementary data if the reviewer insists on such a move but we do not see how that changes the essence of this study.
To provide additional clarification why we tested the double sm1sir1 mutants in the Result section, we added an additional sentence “While our previous research characterized the contrasting functions of the secreted peptides, Sm1 and Sir1, in the regulation of ISR against pathogens, the hypothesized antagonistic interaction between the two signaling peptides was not tested previously”.
How does this result and the work on this pathogen relate to the title and the protection of plants against FAW and WCR?
As we explained in introductory sentences of the relative Results section of the revised version, the reason we have included the pathogen test was because of the results showing that detrimental effect of endophyte treatment on herbivory defense was only evident when plants were grown in sterile soil. This prompted us to test whether there would be a difference in ISR response against C. graminicola, which was only tested in sterile soil in our previously published studies. We believe that the comparison between the soil sterility effect of endophytes impacts on herbivory defense and ISR against pathogen is novel and interesting as these two stressors are rarely compared in the same studies. The future study could be directed at answering the question why we see such a difference.
It is not clear from anything that has been written so far. I thought similar suggestions could be made in all the results sections. So please remove the discussion from the results and avoid repetition, discussion and numerous abbreviations when presenting the results. Also, some of the results point in a different direction than the reader might expect from the title. Does treatment with Trichoderma virens and Pseudomonas chlororaphis protect plants from FAW and/or WCR or not? It was hard to be sure what the right answers was.
We stated in the text that the treatment with both T. virens and P. chlororaphis in sterile soil reduced defense against FAW while did not exhibit any significant change in non-sterile soil. WE also, stated that the treatment of T. virens suppressed WCR larvae survival and weight gain, making it a potential biocontrol agent against this maize pest.
Discussion and Conclusion – When authors made clearly idea about what was the aim(s) in this paper than they should write this two section accordingly. Otherwise at this moment they are both as separate sections.
Because we clarified the Objectives of this study more clearly, now the Discussion should be clearer.